# Validation of suitable genes for normalization of diurnal gene expression studies in *Chenopodium quinoa*

Nathaly Maldonado-Taipe[1], Dilan S. R. Patirange[1], Sandra M. Schmöckel[2,3], Christian Jung[1], Nazgol Emrani[1]*

1 Plant Breeding Institute, Christian-Albrechts-University of Kiel, Kiel, Germany, 2 King Abdullah University of Science and Technology (KAUST), Biological and Environmental Sciences & Engineering Division (BESE), Thuwal, Saudi Arabia, 3 Institute of Crop Science, University of Hohenheim, Stuttgart, Germany

* n.emrani@plantbreeding.uni-kiel.de

**Data Availability Statement:** All relevant data are within the manuscript and its Supporting Information files.

## Abstract

Quinoa depicts high nutritional quality and abiotic stress resistance, attracting strong interest in the last years. To unravel the function of candidate genes for agronomically relevant traits, studying their transcriptional activities by RT-qPCR is an important experimental approach. The accuracy of such experiments strongly depends on precise data normalization. To date, validation of potential candidate genes for normalization of diurnal expression studies has not been performed in *C. quinoa*. We selected eight candidate genes based on transcriptome data and literature survey, including conventionally used reference genes. We used three statistical algorithms (BestKeeper, geNorm and NormFinder) to test their stability and added further validation by a simulation-based strategy. We demonstrated that using different reference genes, including those top ranked by stability, causes significant differences among the resulting diurnal expression patterns. Our results show that isocitrate dehydrogenase enzyme (*IDH-A*) and polypyrimidine tract-binding protein (*PTB*) are suitable genes to normalize diurnal expression data of two different quinoa accessions. Moreover, we validated our reference genes by normalizing two known diurnally regulated genes, *BTC1* and *BBX19*. The validated reference genes obtained in this study will improve the accuracy of RT-qPCR data normalization and facilitate gene expression studies in quinoa.

## 1. Introduction

Real time quantitative PCR (RT-qPCR) is a widely used method, which allows the study of gene expression through quantification of nucleic acids. Normalization of RT-qPCR expression data can be carried out using reference genes. Ideally, the reference gene expression should remain constant or vary minimally in various tissues under the different experimental treatments. However, it has been shown that the utility of the reference genes must be validated for different experimental conditions [1–5], and that normalization against a non-validated reference gene can disprove quantitative results [6]. Screening of an adequate reference gene can be performed by statistical algorithms developed for this purpose. The most popular algorithms are geNorm [7], NormFinder [8] and BestKeeper [9]. geNorm calculates an average

**Funding:** This study was partially funded by the Competitive Research Grant of the King Abdullah University of Science and Technology, Saudi Arabia awarded to CJ (Grant number: OSR-2016-CRG5-2966-02). The funders had no role in study design, data collection and analysis, decision to publish, or preparation of the manuscript. There was no additional external funding received for this study. The rest of the costs were covered internally by the institute budget of the Plant Breeding Institute, Kiel University. We acknowledge the financial support by State of Schleswig-Holstein, Germany within the funding program "Open Access Publikationsfonds".

**Competing interests:** The authors have declared that no competing interests exist.

expression stability, which is defined as the average pairwise variation in a particular gene with all other potential reference genes, whileNormFinder identifies stably expressed genes based on a mathematical model that enables the estimation of the intra and inter-group variation of the sample set, and BestKeeper calculates the standard deviation (SD) and a coefficient of variance (CV) as a measure of stability. Reference genes under different experimental conditions have been found through the use of these algorithms for a variety of crops such as lettuce [10], pea [4], barley [11], peach [3], maize [12], rice [13], pineapple [2] and others.

Many developmental processes are under the influence of the circadian clock. The circadian clock genes supply information on the daily environmental changes and provide plants with the ability to regulate the time of consumption and production of energy. Thus, the circadian system manipulation is a strategy for optimizing plant growth and yield and overcoming environmental stress [14, 15]. The study of the circadian clock by diurnal gene expression through RT-qPCR implies a careful and rigorous selection of reference genes. It has been shown that a reference gene that has been qualified as stable could depict diurnal expression patterns. When such a gene is used to normalize expression data, it causes considerable changes in the expression profiles of the target genes [16].

Quinoa (*Chenopodium quinoa* Willd.) is a dicotyledonous annual species from the family Amaranthaceae. It has an allotetraploid genome (1.39 Gb) presumably originating from the diploid species *Chenopodium pallidicaule* (A-genome) and *Chenopodium suecicum* (B-genome) [17]. Quinoa has high protein content and an exceptional balance between oil, protein and fat. It also supplies balanced amounts of the amino acid lysine along with the other eight essential amino acids [18]. This, together with its great adaptability to extreme climatic and soil conditions, conveys significant potential for quinoa production expansion around the world [19]. To date, no reference genes validation has been performed for diurnal expression studies in *C. quinoa*.

Because normalization against an unsuitable reference gene could produce false expression patterns for the target genes [6], in this study we addressed the question of which reference genes are suitable to normalize diurnal expression data in *C. quinoa*. To answer this question, we selected eight candidate reference genes and used three statistical algorithms (geNorm, NormFinder and BestKeeper) to test their stability. We performed further validation by normalizing the expression data of *DOUBLE B-BOX TYPE ZINC FINGER* (*BBX19*) and *BOLTING TIME CONTROL 1* (*BTC1*), homologs of known diurnally regulated genes that control flowering time regulation in sugar beet [20]. Considering the close phylogenetic relationship between quinoa and sugar beet, we expected these genes to be diurnally regulated in quinoa as well. We normalized the expression of these target genes against each of the candidate reference genes followed by a comparison between the obtained expression profiles and a model expression profile, which we refer to as "simulation". Additionally, we used our selected reference genes to normalize the expression of the photoperiod-response gene *GIGANTEA (GI)*. This work meticulously validated reference genes to facilitate and fortify diurnal gene expression analysis in different accessions of *C. quinoa*.

## 2. Materials and methods

In the present work, we followed the Minimum Information for Publication of Quantitative Real-Time PCR Experiments (MIQE) guidelines [21].

### 2.1 Plant material, growth conditions and tissue sampling

For our main analyses, we considered two *C. quinoa* accessions from different origins: a short day Peruvian accession (CHEN-109; seed code: 170083), which flowered 69 days after sowing

and a Danish accession (Titicaca; seed code: 170191), which is adapted to long day conditions and flowered 38 days after sowing. A third accession from Argentina (PI-587173; seed code: 171605), which flowered 52 days after sowing, was used to prove the broad applicability of the selected reference genes. The seeds were kindly provided by Prof. Mark Tester, King Abdullah University of Science and Technology, Saudi Arabia. One hundred plants per accession were sown in 9 cm pots on February 16th, 2018. The plants grew in growth chamber under short day conditions (8 h light at 22°C, light intensity 120 μE). We performed diurnal sampling of leaves from 5 plants/ accession (A and B samples) every two hours at BBCH 51 (inflorescence emergence stage) [22]. Sampling was performed 24 days after sowing for CHEN-109 and 46 days after sowing for Titicaca. A young leaf sample between 1.0 and 2.0 cm$^2$ was obtained, placed in a 2 ml Eppendorf tube and immersed into liquid nitrogen. Samples were kept at -70°C until RNA extraction.

## 2.2 RNA isolation and cDNA synthesis

From a total of 13 sampling points described in the previous section, we isolated RNA from seven time points. Total RNA was extracted from three biological replicates of each accession for the Zeitgeber points ZT-0, ZT-4, ZT-8, ZT-12, ZT-16, ZT-20 and ZT-24 using the NucleoSpin RNA Plant kit (Macherey-Nagel, Düren, Germany). We performed a DNase I treatment (Thermo Fisher Scientific Inc., Waltham, United States) for 30 min at 37°C on the isolated RNA to eliminate DNA contamination. To verify that DNA was removed, we carried out a standard PCR (annealing temperature: 58°C; 40 cycles) for the DNase treated RNA with specific primers designed in intronic regions of *BOLTING TIME CONTROL 1* (*BTC1*) (Gene ID: *AUR62041615;* Fw: GGAAGAAATCAAGGGTATTGACTCTAGAG, Rv: GAACCATATACAAGC CTTTCTGAGAG). We controlled the RNA integrity through agarose gel electrophoresis (1.5%) and RNA concentration and purity were measured with a NanoDrop 2000 Spectrophotometer (Thermo Fisher Scientific Inc., Waltham, United States). An $A_{260}/A_{280}$ ratio between 2.0 and 2.2 and an $A_{260}/A_{230}$ ratio higher than 1.7 indicated appropriate RNA quality. Exemplary RNA agarose gels and the $A_{260}/A_{280}$ and $A_{260}/A_{230}$ ratios for the Titicaca accession are presented in S1 Fig and S1 Table, respectively. We used the DNase-treated RNA to perform the reverse transcription with the First Strand cDNA Synthesis Kit (Thermo Fisher Scientific Inc., Waltham, United States), according to the manufacturer's instructions. We used the RNA concentration obtained during the NanoDrop analysis to calculate the required volume of every sample to load a final amount of 1 ug of RNA to the reverse transcription process. Later, we used the products of the reverse transcription to perform a standard PCR (annealing temperature: 56°C; 36 cycles; *RAN-3* primers in Table 1) followed by agarose gel electrophoresis (2.0%) to verify the success of the cDNA synthesis (S1 Fig). The synthesized cDNA was stored at -20°C for further use.

## 2.3 Selection of candidate reference genes

We made a literature survey to select potential candidate reference genes. The most commonly used reference genes, preferably the ones that have been previously reported to be efficient in species of the Amaranthacea family, were selected [23, 24]. Afterwards, we validated this list of genes based on the available *C. quinoa* transcriptome data [17] by selecting those genes, which were stably expressed at least in two out of four stress conditions (low phosphate, salt, drought and heat) for soil- and hydroponic-grown plants within root and shoot tissues (one-way ANOVA and Tukey's Multiple Comparison Test; α = 0.05) (S2 Table).

We designed paralog-specific primer pairs for amplification of the candidate reference genes except for *GAPDH*, whose primers were designed in a paralog-conserved region. We

**Table 1. Primer sequences, amplification product size and efficiency of the selected candidate reference genes and the target genes.**

| Gene ID | Gene description | Gene abbreviation | Primer sequence 5'-3' (bp) | Amplification product size (bp) | Amplification efficiency (%) | Regression coefficient (R²) | Tm (°C) |
|---|---|---|---|---|---|---|---|
| AUR62005573 | ACTIN-11 | ACT | Fw-GCTGGAAGGTGCTCAGAGATGC | 150 | 92.0 | 0.925 | 60 |
| | | | Rv-CGGCGATACCAGGGAACATGG | | | | |
| AUR62005566 | GLYCERALDEHYDE-3-PHOSPHATE DEHYDROGENASE, CYTOSOLIC | GAPDH* | Fw-CGGCTTCCTTCAACATCATTCCTAGC | 144 | 85.5 | 0.936 | 60 |
| AUR62024167 | | | Rv-CGGTTGTTGATCTCACTGTCAGGC | | | | |
| AUR62030674 | GUANOSINE TRIPHOSPHATE BINDING NUCLEAR PROTEIN RAN-3 | RAN-3* | Fw-CTTTGAGAAGCCCTTCCTCTAC | 135 | 107.0 | 0.969 | 56 |
| | | | Rv-AGCTCTTGTTCATGCCTCTG | | | | |
| AUR62002238 | ISOCITRATE DEHYDROGENASE | IDH-A | Fw-GGACGGACTATTGAAGCTGAAGC | 125 | 80.8 | 0.996 | 60 |
| | | | Rv-GCATGGACACGTGGGCTTGC | | | | |
| AUR62037769 | ISOCITRATE DEHYDROGENASE | IDH-B* | Fw-GATCTTGGTCTTCCCCACCG | 164 | 86.8 | 0.986 | 57 |
| | | | Rv-GCAAATGTGGAAAAGCCCAAATGG | | | | |
| AUR6203443 | POLYPYRIMIDINE TRACT-BINDING PROTEIN | PTB | Fw-GCAGTGCAACAGGCTCCAAG | 143 | 70.0 | 0.987 | 57 |
| | | | Rv-GGCACCTGCACCCTCATG | | | | |
| AUR62007961 | TUBULIN ALPHA-1 CHAIN | TUB | Fw-ACTCCACCAGTGTTGCTGAG | 115 | 132.1 | 0.958 | 57 |
| | | | Rv-TCAAACCCTTCCTCCATACC | | | | |
| AUR62015654 | POLYUBIQUITIN 10 | UBQ | Fw-ATCTGGTGCTCCGTTTGAG | 114 | 79.4 | 0.956 | 57 |
| | | | Rv-TCTTCGCCTTAACATTGTCG | | | | |
| AUR62041615 | DOUBLE B-BOX TYPE ZINC FINGER | BBX19* | Fw-GAGAAGGAGCAAAATCATGTAG | 158 | 71.7 | 0.984 | 55 |
| AUR62036007 | | | Rv-GAATGGACTATGTTCCTGGAAC | | | | |
| AUR62041615 | BOLTING TIME CONTROL 1 | BTC1 | Fw-GCAGCAAAGCAATCCAATTCTGAC | 151 | 85.1 | 0.945 | 58 |
| AUR62036007 | | | Rv-GCAAAATCAACAGTCCATGGTTCTC | | | | |
| AUR62034894 | GIGANTEA | GI* | Fw-ATGGGGAGAGTCTGGATTAG | 152 | 101.2 | 0.961 | 60 |
| | | | Rv-CAGATCCGTTCTGCTGTATAT | | | | |

Gene IDs are described according to KAUST Repository: http://www.cbrc.kaust.edu.sa/chenopodiumdb/. Uppercase -A/-B letters in the gene abbreviation describe sub-genome location. Tm = melting temperature. Asterisks show genes for which the primers were designed in different exons.

designed primers in a paralog-conserved region for the target genes *BBX19* and *BTC1*, while for *GI*, paralog-specific primers were designed (Table 1). We assessed the primer quality with the OligoCalc program (http://biotools.nubic.northwestern.edu/OligoCalc.html). We considered the following primer quality parameters: length (18–26 bp), similar melting temperature (50–60°C), amplicon length (100–160 bp), GC content (>40%) and lack of self-annealing and primer-dimmer formation. To design the *GAPDH* primers, a Multiple Sequence Alignment of *GAPDH* paralogs with CLC Main Workbench (QIAGEN Aarhus, Denmark) was performed with the following parameters; gap open cost: 10, gap extension cost: 1, end gap cost: as any other and very accurate alignment. Whenever it was possible, we designed primers across introns and checked the lack of gDNA amplification by verification of the expected amplification size of the cDNA (Table 1). We verified the specificity of the primers by standard PCR using DNA from Titicaca accession followed by agarose gel electrophoresis. Moreover, we sent the products of locus-specific primers for Sanger sequencing to the Institute for Clinical Molecular Biology (IKMB, Kiel University) and the sequencing results were analyzed with CLC Main Workbench. A further verification of the primer combinations specificity was

made by examination of the RT-qPCR melting curves using the Bio-Rad CFX Manager 3.1 software (Bio-Rad Laboratories GmbH, Munich, Germany).

## 2.4 Quantitative real-time PCR (RT-qPCR) and determination of amplification efficiency

We performed RT-qPCR with a Bio-Rad CFX96 Real-Time System, which has a built-in Bio-Rad C1000 Thermal Cycler (Bio-Rad Laboratories GmbH, Munich, Germany) using Platinum SYBR Green qPCR SuperMix-UDG with ROX (Invitrogen by Life Technologies GmbH, Darmstadt, Germany). Every PCR reaction had a total volume of 20 μl:10 μl of Platinum SYBR Green qPCR SuperMix-UDG, 1 μl of each forward and reverse primer (10 μg), 2 μl of diluted cDNA (1:20 dilution) and 6 μl of ddH$_2$O. The amplification conditions were as follows: 95°C for 3 min as initial denaturation and 40 cycles of: 10 s at 95°C, 20 s at primer pair annealing temperature and 30 s at 72°C. Recording of melting curves was performed from 60°C to 95°C every 0.5°C to confirm primer specificity and lack of primer dimmers. Three technical replicates per sample and per negative control (water) were also loaded. We amplified cDNA dilution series (dilution factors 1:40 (twice), 1:20, 1:10 and 3:20) to generate standard curves for estimation of the correlation coefficient (R$^2$) and the amplification efficiency $E = (10^{-\frac{1}{slope}} - 1) \times 100$. The correlation coefficient (R$^2$) is given for the regression line of the log of the starting quantity (dilution determined; x axis) and the corresponding Cq value (y axis). The standard curves for the reference genes and the target genes *BBX19*, *BTC1* and *GI* can be found in S2 Fig. Cq values were obtained by setting the baseline threshold to 100. We used the Bio-Rad CFX Manager 3.1 software (Bio-Rad Laboratories GmbH, Munich, Germany).

## 2.5 Gene stability analysis

We assessed the expression stability of the candidate genes by three algorithms: geNorm [7], NormFinder [8] and BestKeeper [9]. The data used for these analyses were the averaged Cq value of the three biological replicates (three technical replicates each) obtained for seven different Zeitgeber points. For the determination of the stability by geNorm and NormFinder, transformation of the raw Cq values into relative quantities (RQ) was performed as follows:

$$\mathbf{RQ} = \mathbf{E}^{-\Delta \mathbf{Cq}}$$

where, E represents the PCR efficiency and ΔCq for any given gene is the Cq value of each sample subtracted from the lowest Cq value among all the samples for that gene. In contrast, for the utilization of BestKeeper, the raw non-transformed Cq data were used, as required by the algorithm.

The geNorm algorithm calculates an average expression stability, M-value, which is defined as the average pairwise variation in a particular gene with all other potential reference genes. This software was also used to calculate the minimum number of genes required for normalization, which is given by the pairwise variation as follows:

$$\boldsymbol{V_n/V_{n+1}}$$

where, $V_n$ is the pairwise variation of a given number of reference genes and $V_{n+1}$ is the pairwise variation produced when an extra reference gene is added to the analysis. Below the default cut-off value of 0.15, all gene pairs were considered stable and the addition of an extra gene for normalization was not required [7]. NormFinder identifies stably expressed genes based on a mathematical model that enables the estimation of the intra and inter-group

variation of the sample set as a stability value (SV) [8]. In order to obtain the best combination of reference genes together with its stability value and the intra and intergroup variations, we added group identifiers to distinguish the expression data of CHEN-109 and Titicaca. In the case of BestKeeper, it helps in selection of the best reference genes after the calculation of the standard deviation (SD) and a coefficient of variance (CV) [9]. Following the algorithms protocols, we considered candidate reference genes with the lowest M, SV and CV values as the most stably expressed genes.

## 2.6 Validation of the reference genes

To validate the reliability of the top ranked reference genes and to clarify the ambiguous results given by the algorithms, we normalized the expression data of the diurnally regulated genes *BBX19* and *BTC1* [20] against each of the candidate genes. Additionally, we simulated the expected expression patterns of *CqBBX19* and *CqBTC1*. In the simulation, the Cq values of the target genes were normalized against a constant Cq value of 20, which represents an ideal reference gene. In other words, the Cq value of the candidate reference genes is replaced by a constant value of 20 for all sampling points and biological replicates for the purpose of simulation. Then, a comparison between the simulation and the obtained expression profiles of the target genes was performed. A reference gene was not considered as suitable, when its normalization results differed from the simulation. Number and position of peaks and the change of the expression values between accessions (e.g., when an accession has a higher expression value compared to the other in the simulation, the same feature should be depicted during the actual normalization) were examined. To detect the differences between the obtained expression profiles and the simulation, we performed t-tests ($\alpha = 0.05$) at every Zeitgeber point between the accessions and one-way ANOVA ($\alpha = 0.05$) for the number and position of peaks. To further validate the broad suitability of the reference genes, we repeated the same procedure for another target gene, *GI*, in the accession PI-587173. In this way, one could expect an assortment of diurnal patterns in the investigated target genes: a low peak at the down for *BBX19* [20], a high morning peak for *BTC1* [25], and a sharp high peak at dawn for *GI* [26].

Expression levels for all accessions were determined by $2^{-\Delta\Delta Cq}$ method [27]. We designed the primer combinations for *CqBBX19*, *CqBTC1* and *CqGI* (Table 1) according to the criteria mentioned above and their specificity was assessed in the same way as the previously described for the reference genes primer pairs.

## 3. Results

### 3.1 Selection of candidate genes

First, we performed a selection of candidate genes based on a literature survey. We preferentially selected reference genes that have been validated in species of the Amaranthacea family [23, 24]. Afterwards, we validated these genes with *C. quinoa* transcriptome data, where plants had been grown under abiotic stress conditions [17]. We selected eight candidate reference genes (Table 1), which were stably expressed in root and shoot tissues in at least two out of four stress conditions (S2 Table).

Then, we designed primer combinations and assessed their specificity by standard PCR, Sanger sequencing and melting curve analysis. Out of 23 primer combinations, 16 primer pairs were selected. The specificity of the primer combinations was demonstrated by the presence of a single reaction product of the expected size (S3 Fig), the presence of a single peak in the RT-qPCR melting curve (S4 Fig), and an alignment of the sequenced products to the reference sequence showing one peak per nucleotide (S5 Fig). Amplification efficiencies of the

primer combinations varied between 70.0 and 132.1% with regression coefficients ($R^2$) varying from 0.925 to 0.996 (Table 1).

## 3.2 Diurnal expression profiles of candidate genes

In a next step, we determined the expression profiles of the candidate genes by RT-qPCR and the raw Cq values were used to quantify the expression levels. We choose two accessions from Peru (CHEN-109) and Denmark (Titicaca), because of their different photoperiod sensitivity, which might be related to their geographical origins [28]. We extracted RNA from samples taken during one day at seven different time points for each accession. The expression range across all samples of the eight candidate genes was of $21.23 \leq Cq \leq 27.98$ for CHEN-109 and of $21.19 \leq Cq \leq 28.01$ for Titicaca (Fig 1). Unexpectedly, the expression profiles of some of the candidate genes across the samples showed clear diurnal patterns for both accessions (Fig 2), implying a low transcript stability over the day. For instance, in Titicaca *IDH-B*, *GAPDH* and *ACT* depicted a peak in their expression pattern at ZT-20 (one-way ANOVA; α = 0.05). Contrarily, in CHEN-109 the transcriptional activities of these genes peaked at different time points (*IDH-B* at ZT-8) or displayed no diurnal regulation (e.g. *GAPDH*) (one-way ANOVA; α = 0.05).

## 3.3 Expression stability of the candidate genes in two quinoa accessions

We assessed the stability of the candidate genes by three algorithms: geNorm [7], NormFinder [8] and BestKeeper [9]. geNorm and BestKeeper placed *PTB* and *IDH-A* as the most stable

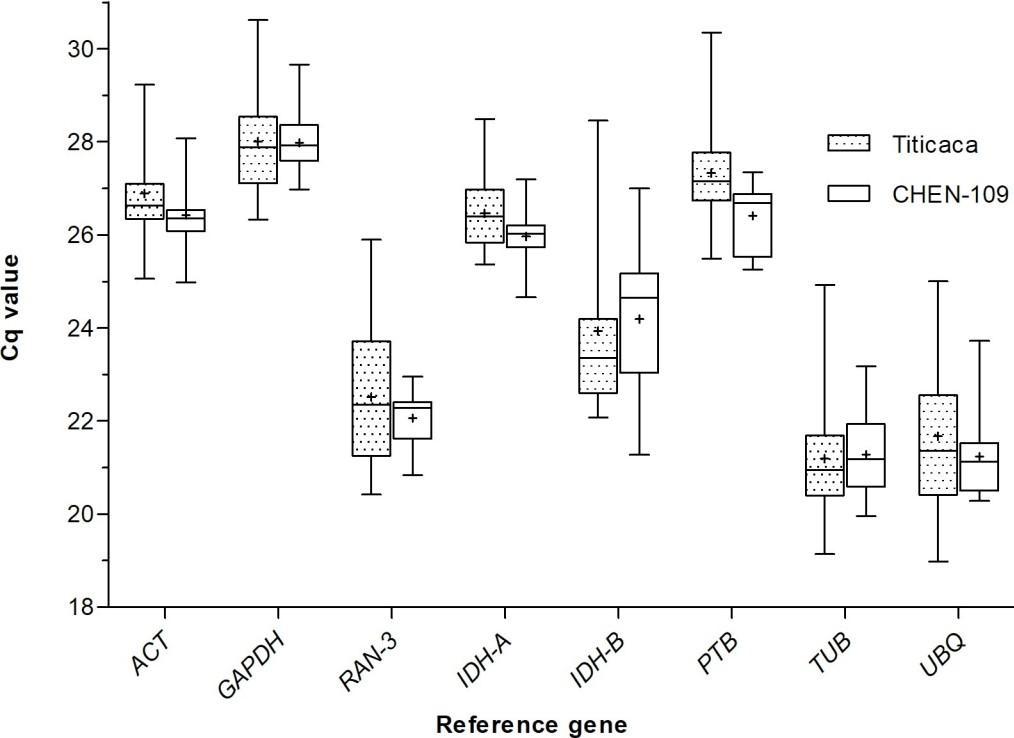

**Fig 1. Variation in Cq values of the candidate genes.** Cq values were calculated as means of triplicate technical replicates across the 21 diurnal/circadian samples. The median values are represented as lines across the box. The lower and the upper boundaries of each box represent the 25th and 75th percentile, respectively. Whiskers represent the maximum and minimum values. The mean is represented by crosses inside the boxes.

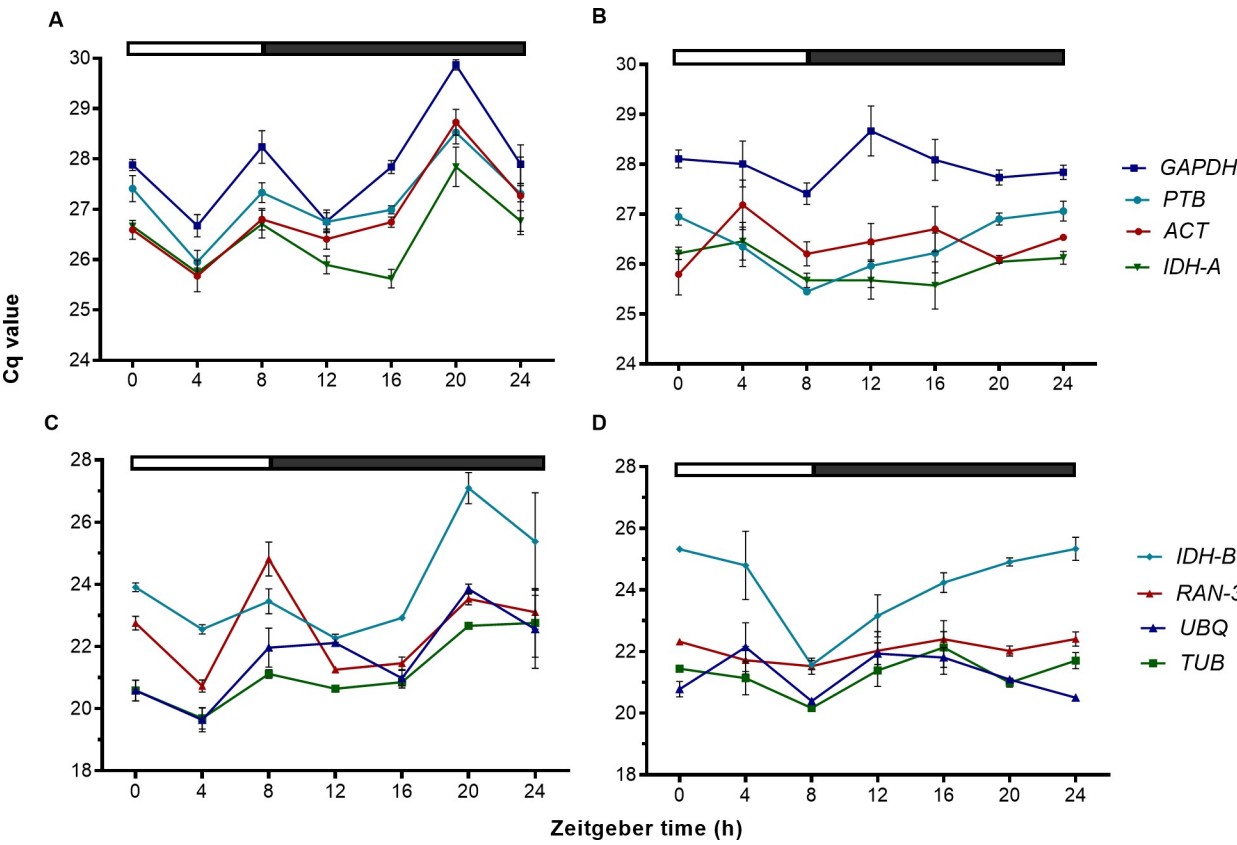

**Fig 2. Expression profiles of candidate genes.** Cq values from 21 samples (three technical replicates of three biological replicates) taken at seven different points of the day for (A,C) Titicaca and (B,D) CHEN-109. Two groups of genes are represented: (A,B) high Cq value and (C,D) low Cq value. The bar at the top indicates light (empty box) and dark (filled box) phases. The error bars represent the SEM between biological replicates.

genes for normalization of Titicaca expression data. On the other hand, the same algorithms placed *TUB* and *IDH-A* as the most stable genes for normalization of CHEN-109 expression data (Fig 3). NormFinder determined the most suitable reference gene for the two sets of expression data, simultaneously (Titicaca and CHEN-109), ranking *ACT* as the most suitable single gene and *IDH-A* together with *TUB* as the best combination of genes (stability value 0.035) (Fig 3 and S6 Fig). Additionally, we performed a pairwise variation analysis (cut-off value of 0.15) of the candidate genes for determination of the appropriate number of reference genes by geNorm. The appropriate number of genes required for normalization in this study was two (Fig 4).

## 3.4 Validation of selected reference genes for normalization of diurnal expression data

We performed a further validation of the candidate genes due to their diurnal expression patterns (Fig 2) and their dissimilarity in the rankings obtained from the algorithms (Fig 3). We used the diurnally regulated genes *CqBBX19* and *CqBTC1* that share sequence homology with *BvBBX19* and *BvBTC1* respectively. These are major flowering time regulators of sugar beet (*Beta vulgaris* L.) [20] and because both species belong to the same plant family, it is tempting to speculate that *CqBBX19* and *CqBTC1* have a similar function in quinoa. We normalized the expression data of *CqBBX19* (Fig 5) and *CqBTC1* (Fig 6) against each of the candidate

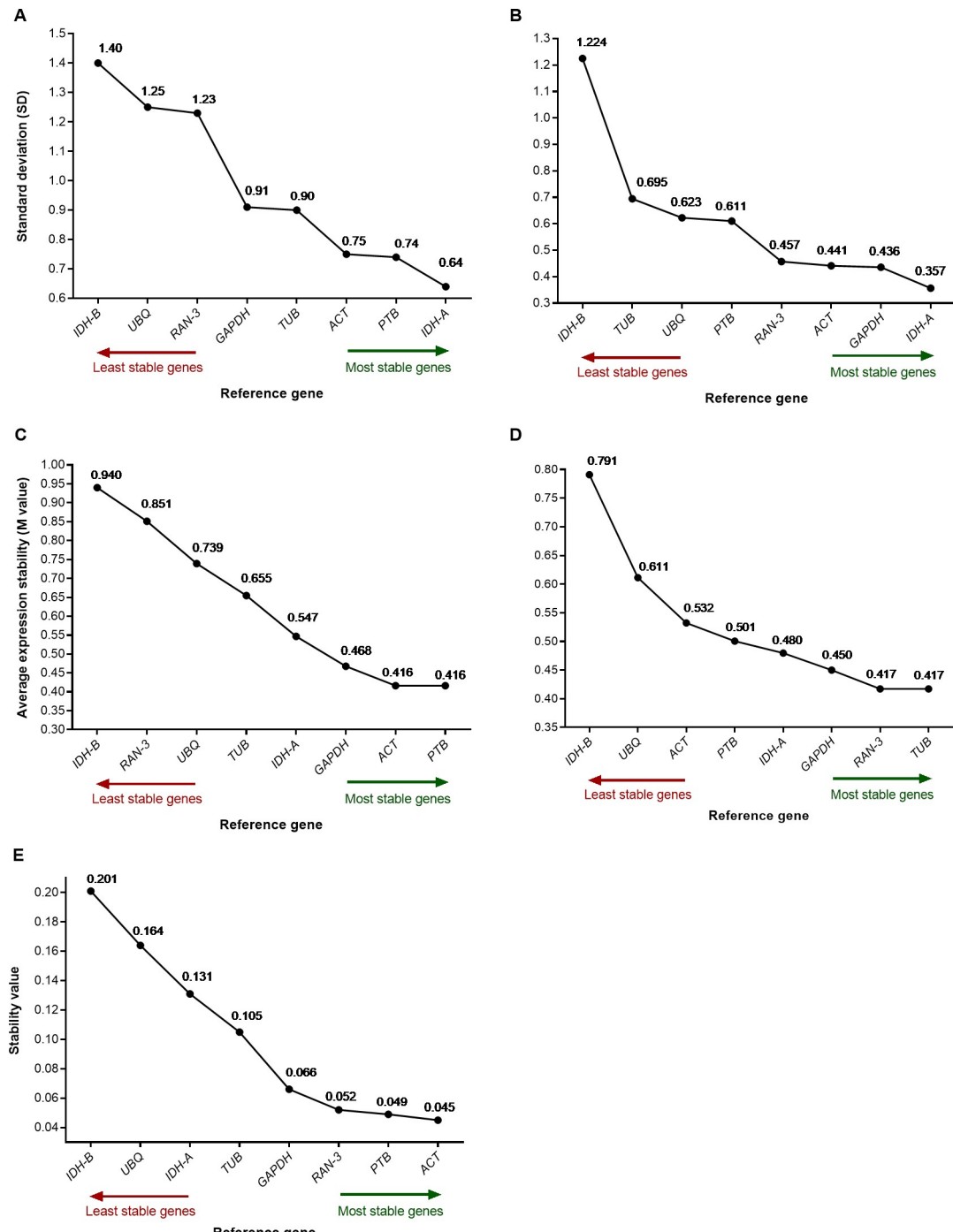

**Fig 3. Expression stability ranking of the candidate genes.** Rankings were calculated by (A,B) BestKeeper, (C,D) geNorm and (E) NormFinder for two *C. quinoa* accessions: (A,C) Titicaca and (B,D) CHEN-109. Stability values, SD and M values are shown for every gene. The most stable genes are shown on the right side (green arrow) and the least stable genes on the left (red arrow).

reference genes. Furthermore, we carried out the normalization of both target genes with a constant Cq value of 20, to identify the expected expression patterns (simulation). Then, we looked for critical differences between the simulation and the obtained expression profiles of

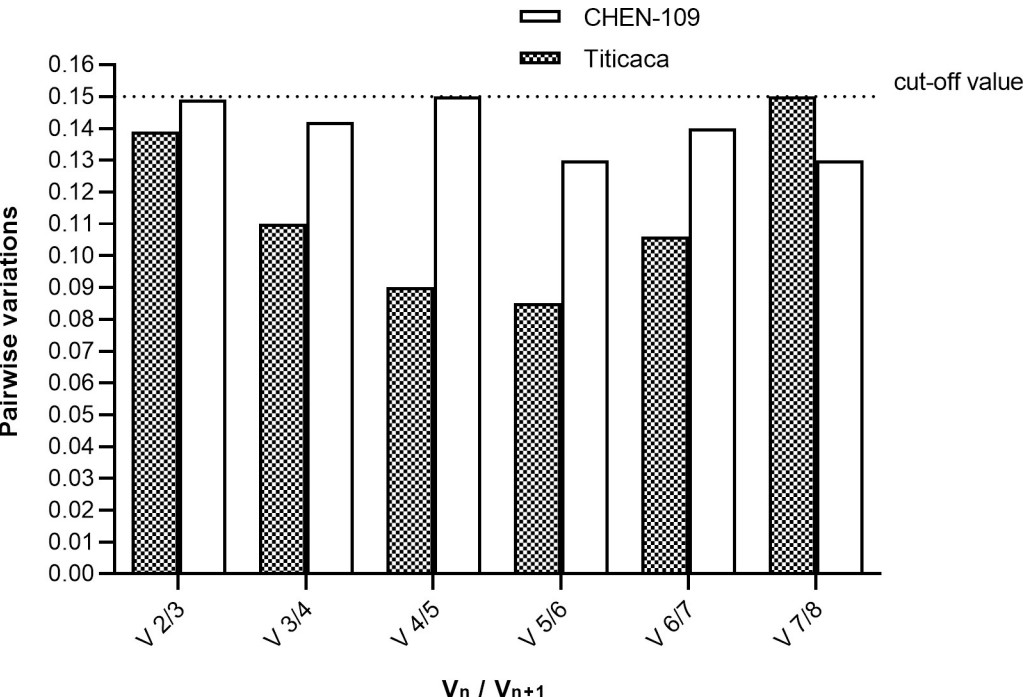

**Fig 4. Pairwise variation analysis of the candidate genes.** The analysis was performed by geNorm for determination of the appropriate number of reference genes. Below 0.15 (default cut-off value) all gene pairs were considered stable and the addition of a third gene for normalization is not required. $V_n$ is the pairwise variation of a given number of reference genes and $V_{n+1}$ is the pairwise variation produced when an extra reference gene is added to the analysis.

the target genes (e.g. number and position of expression peaks) and statistically assessed the significance of these differences.

As an outcome, normalizing *CqBBX19* expression data against *GAPDH*, *IDH-B*, *UBQ*, *TUB*, *ACT* and *RAN-3* resulted in expression patterns that clearly differed from the simulation in number, shape and position of the expression peaks (Fig 5), which suggests that these genes are unsuitable references and may be discarded. For instance, expression patterns obtained for CHEN-109 by normalization against *UBQ* and *ACT* depicted a peak at ZT-4, which is absent in the simulated expression pattern (one-way ANOVA; $\alpha = 0.05$) and constitutes a great distortion from the expected pattern predicted by the simulation. On the other hand, the combination of the two best genes for normalization in Titicaca determined by BestKeeper (*IDH-A* + *PTB*) exhibited analogous diurnal expression patterns to the simulation (one-way ANOVA; $\alpha = 0.05$). Similar observations can be made in the case of *CqBTC1*, where for instance, normalization against *IDH-B* or *UBQ* resulted in a peak at ZT-4 (one-way ANOVA; $\alpha = 0.05$). This unique peak is not only absent in the simulation, but also in the expression profiles obtained by normalization against other reference genes (Fig 6).

We concluded that *IDH-A* and *PTB* geometric mean is the best reference to normalize diurnal expression studies in *C. quinoa*. The results of GeNorm indicated *IDH-A* and *PTB* as the best references and showed that the geometric mean of two reference genes is required for accurate normalization. This, taken together with the simulation outcome, in which expression results normalized against *IDH-A* and *PTB* did not differ from the simulation of candidate genes for both accessions, brought us to this conclusion. To provide more evidence of the suitability of the chosen reference, we studied the relative expression of the circadian gene *CqGI* in PI-587173 (S7 Fig). The simulation predicted a peak of expression at ZT-4, which is extended

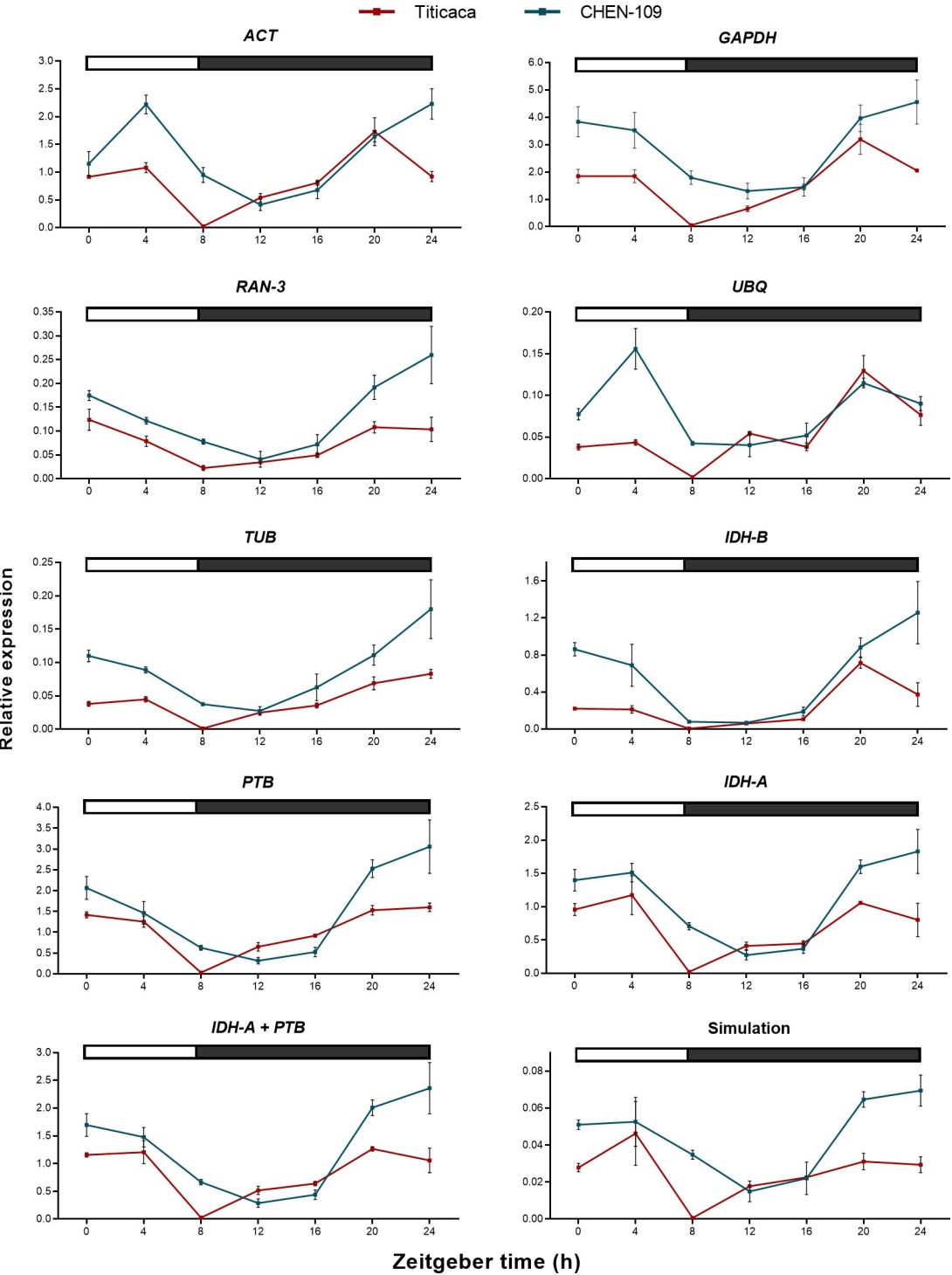

**Fig 5. Validation of reference genes using the target gene *CqBBX19*.** Relative *CqBBX19* expression normalized against each candidate gene, the combination of the two best genes for normalization of Titicaca determined by BestKeeper (*IDH-A + PTB*) and a constant Cq value of 20 (Simulation). The bar at the top indicates light (empty box) and dark (filled box) phases. Error bars represent the SEM of three biological replicates.

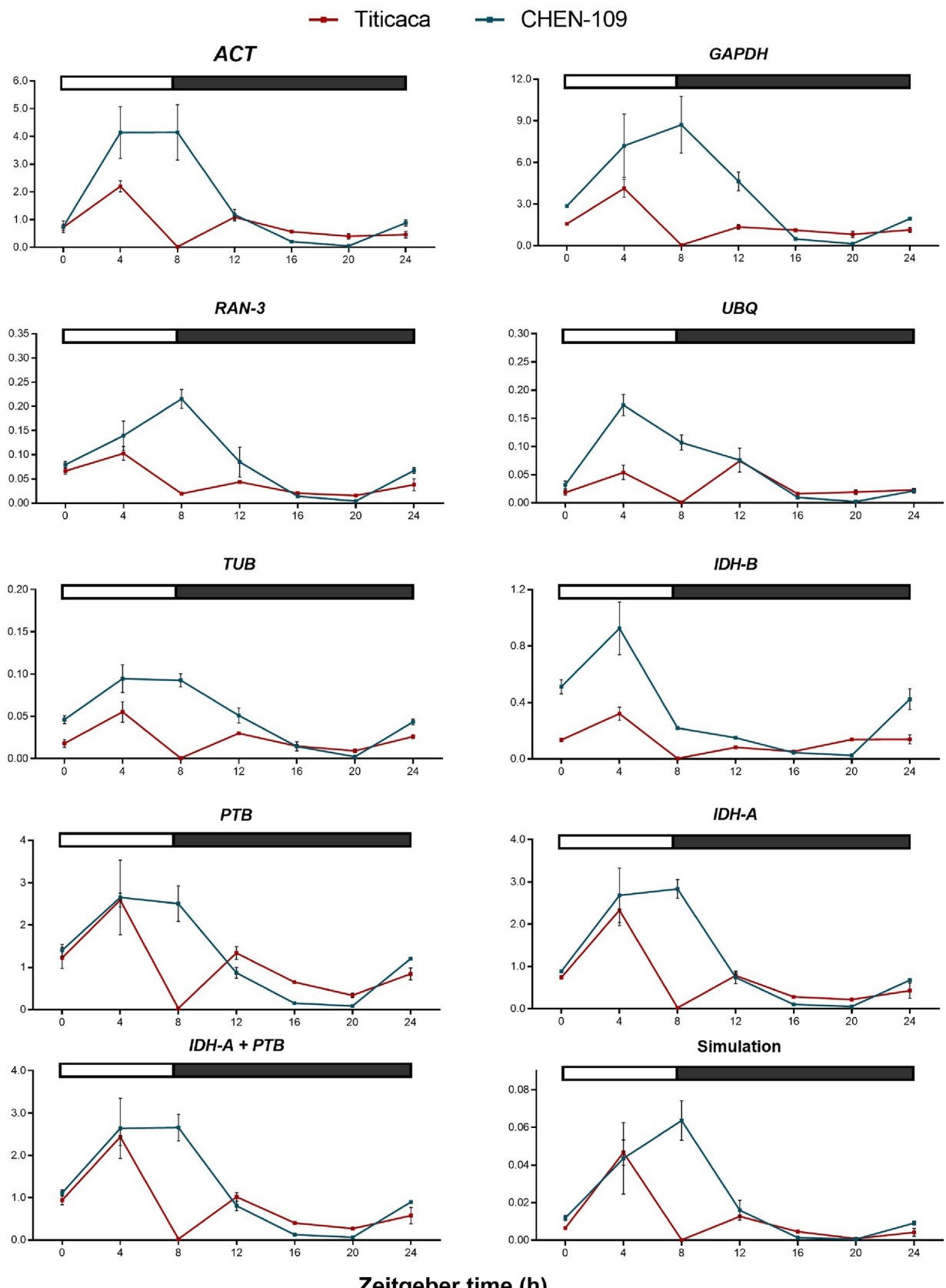

**Fig 6. Validation of reference genes using the target gene *CqBTC1*.** Relative *CqBTC1* expression normalized against each candidate gene, the combination of the two best genes for normalization of Titicaca determined by BestKeeper (*IDH-A* + *PTB*) and a constant Cq value of 20 (Simulation). The bar at the top indicates light (empty box) and dark (filled box) phases. Error bars represent the SEM of three biological replicates.

until ZT-8 and drops at ZT-12 (one-way ANOVA; $\alpha = 0.05$). This predicted pattern was similar to the expression pattern of *CqGI*, when normalized against *IDH-A* and *PTB* geometric mean, showing the appropriate choice of the reference genes.

Our results show that using different reference genes for normalization, including those top ranked by stability, causes significant differences among the diurnal expression patterns of *CqBBX19*, *CqBTC1* and *CqGI* under short day conditions. We were able to find a suitable reference for the normalization of our diurnal expression data by adding a simulation analysis to the data derived from commonly used software packages, avoiding selection of reference genes that produce false transcriptional profiles.

## 4. Discussion

RT-qPCR is a powerful tool to study gene expression, which allows the measurement of small amounts of nucleic acids in a wide range of samples from numerous sources. Nevertheless, the accuracy of RT-qPCR results strongly depends on accurate transcript normalization, which is very frequently overlooked during the data assessment. Moreover, misled interpretations guided by inaccurate results have been reported by using a single reference with no evidence that it is stably expressed across all conditions [6]. Furthermore, finding a gene that is stably expressed through a 24 h cycle represents a challenge, because transcript abundance of thousands of genes interplay during the day and conventionally used reference genes may show cyclic expression patterns [16]. In this study, we aimed to select suitable reference genes for diurnal expression studies to normalize expression data from two different quinoa accessions. We selected eight candidate reference genes (*ACT*, *GAPDH*, *RAN-3*, *IDH-A*, *IDH-B*, *PTB*, *UBQ*, *TUB*) based on *C. quinoa* transcriptome data [17] and literature survey [16, 23, 24] and tested them for stability during the course of a day. By adding a simulation analysis to the data obtained from conventional software packages (geNorm, NormFinder and BestKeeper), we determined that a combination of *IDH-A* and *PTB* was the most suitable reference to normalize our diurnal expression data. To date, this is the first report of a systematic analysis of reference genes for normalization of gene expression data in diurnal studies in *C. quinoa*.

Housekeeping genes are typically used as reference genes because their transcript levels are assumed stable during the execution of basic cellular functions. Nevertheless, the circadian regulatory pathways involve complex and intricate processes [14, 29], in which a housekeeping gene can undergo circadian regulation during the execution of its housekeeping function or the execution of a different function coupled to a circadian pathway (e.g. light induced *GAPDH* response in corn seedlings [30] and *TUB* carbon-induced diurnal response in Arabidopsis rosettes [31]). Furthermore, the complexity of the circadian pathways is reflected in the total number of diurnally expressed genes in plants, which varies widely from species to species. For instance, the percentage of rhythmic genes in transcriptome datasets of Arabidopsis rosettes and rice seedlings is 37.1% (14,019) and 33.2% (24,957), respectively [32]. In addition, the number of diurnally expressed genes highly depends on the tissue and developmental stage [32, 33]. Therefore, it is expected that conventionally used reference genes depict cyclic regulation, as it has been already illustrated for rice plants subjected to circadian cycles of different temperature and day length conditions [16]. Additionally, the observed differences in the expression profiles between Titicaca and CHEN-109 may be due to the origin-related

photoperiod sensitivity of the accessions [28]. Similar intra-species variations, where a gene which is stable in one experiment is unstable in another one, has been observed in peach [3], rice [13] and petunia [34]. Ultimately, the described differences between accessions together with the observed diurnal pattern of the expression profiles enhance the need for a rigorous expression stability analysis. Moreover, when the validation of the reference genes is well performed, the use of suitable reference genes reduces experimentally induced or inherent technical variations [6]. In this sense, primer selection is a relevant point to consider, since several parameters concerning primer development can affect the RT-qPCR efficiency [21]. In our study, primer specificity was carefully assessed. Additionally, the RNA quality controls declined the possibility of samples containing PCR inhibitors. The selection of multiple-intron spanning primers is preferred, because it can prevent amplification of gDNA. In our case, this was not possible for every candidate gene due to other considered primer development criteria. However, we eliminated the possibility of gDNA amplification by performing a test PCR with our DNase treated RNA samples prior to cDNA synthesis. We only considered samples for which no amplification was detected using RNA as template, which would indicate absence of gDNA contamination.

From the analysis of the stability of the candidate reference genes, we obtained contrasting results using geNorm, NormFinder and BestKeeper. The variations we observed in the rankings across all the algorithms are explicable as they use different statistical approaches. Ranking the reference genes by pairwise comparison approaches (geNorm and BestKeeper) is problematic, since some genes may have similar but not constant expression profiles and will be ranked as the most stable reference genes irrespective of their expression stability [8]. The model-based algorithm NormFinder intuitively adds two sources of variation (intra and intergroup). When two different groups depict different expression profiles for the same candidate, the intergroup variation may mask the intragroup variation. Consequently, a suitable reference gene could be poorly ranked despite of presenting a low variation within groups. In this study, for instance, the expression profile of *IDH-A* was different for CHEN-109 and Titicaca and occupied the position six in the ranking despite of its low variation within accessions. Several studies have identified negligible to major differences in the stability rankings of the same gene. RefFinder and RankAgreeg are commonly used tools to configure a standard ranking for reference genes [3, 5, 11]. Here, we introduced a novel and straightforward method to elucidate the ambiguous results of the commonly used software packages in order to find suitable reference genes for normalization of diurnal expression data. The approach is based on the normalization of the target genes expression data against a constant Cq value of 20, which simulates the "ideal" reference gene with constant expression levels unaffected by experimental factors [21]. A comparison between the simulated results and the normalized expressions against the different candidate reference genes could be used to identify distorted diurnal patterns such as shifted peaks. Thus, the simulation is particularly useful to disregard inefficient reference genes. This approach completes the stability-based selection of reference genes based on geNorm, NormFinder and BestKeeper. The ideal reference gene follows these principles: the expression level is similar to that of the target gene, gene is expressed to a certain degree and its expression is not affected by any experimental conditions [35–37]. Thus, replacing the Cq values of the reference gene by a constant Cq value, mimics the ideal reference gene expression. Changing the selected constant Cq value for the simulation considering these principles will not affect the simulation pattern since the expression is reported as relative values (S8 Fig). It is appropriate to mention that the simulation-based approach works under the assumption that identical amount of cDNA was added to each qPCR reaction. Thus, the method relies on accurate measurement of the input to the RT-qPCR reaction. Although equal concentration of RNA for all the samples was considered for cDNA synthesis, DNase treatment and reverse

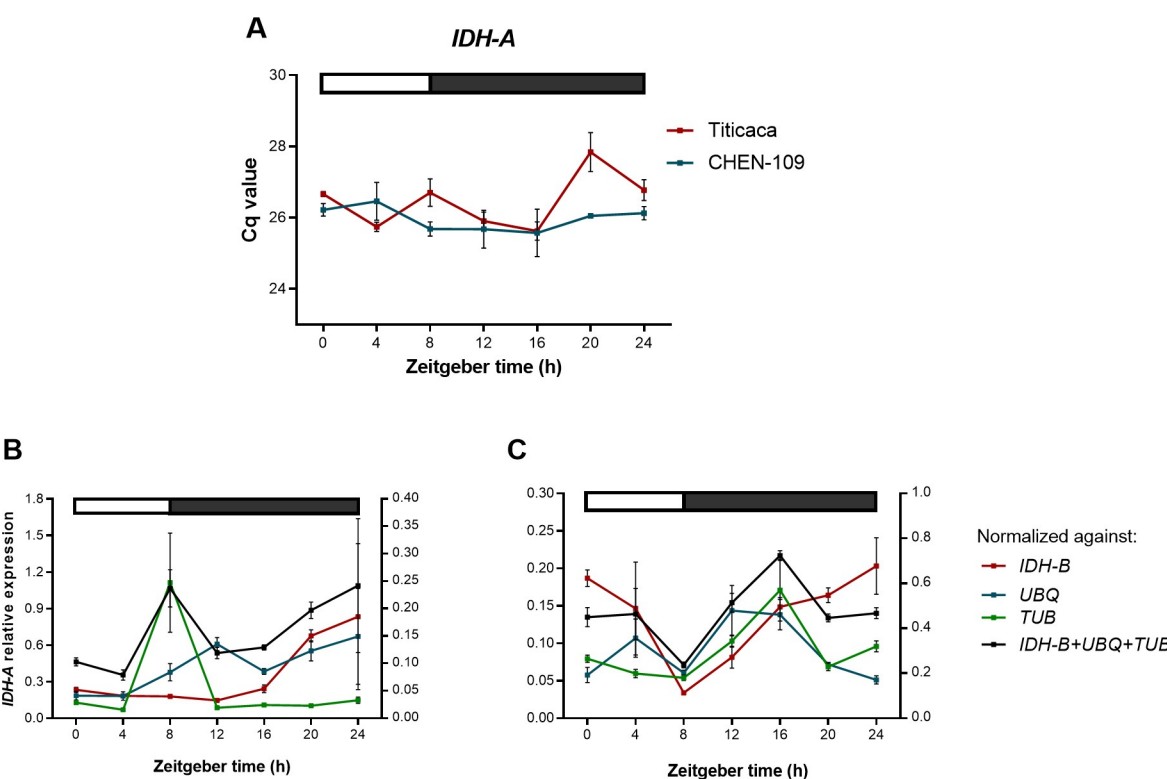

**Fig 7. Illustration of a non-diurnal gene misrepresented as diurnally regulated.** (A) Expression profile of *IDH-A* (best reference gene according to BestKeeper), which did not show cyclic expression; error bars represent standard error between biological replicates. Relative *IDH-A* expression normalized with each of the three least stable reference genes according to BestKeeper and their geometric mean for (B) Titicaca and (C) CHEN-109. Error bars represent the SEM between biological replicates. The bar at the top indicates light (empty box) and dark (filled box) phases.

transcription may generate tube-to-tube differences in the final cDNA yield. We relied on Nanodrop measurements of RNA before performing reverse transcription and fulfilled the MIQE guidelines (S3 Table) [21]. Nevertheless, a fluorescent-based method to measure RNA concentration and, better yet, cDNA concentration, is advisable for future use of the simulation-based approach [38].

In the current study, commonly used reference genes were considered unstable because they resulted in diurnal expression patterns of the target genes, which differed from the simulated pattern. Moreover, reference genes selected based on quinoa transcriptome data[18] showed unstable expression pattern in our dataset, which emphasizes the fact that they should be further validated for different environmental conditions. *GAPDH*, *ACT*, *UBQ*, *RAN-3* and *TUB* have been already shown to be unsuitable as reference genes under certain experimental conditions [2, 4, 16]. The lack of stability of some of the widely used reference genes could not be detected by the algorithms in diurnal studies in rice [16]. Moreover, the use of inadequate reference genes could result in the representation of a gene as diurnally regulated, even when the gene does not depict a diurnal expression pattern in reality. To illustrate this, we took *IDH-A* expression data and treated it as target gene. We selected *IDH-A* because it depicts a stable expression profile (Fig 7A) and we do not expect it to be diurnally regulated. Then we used the three least stable reference genes (BestKeeper) and their geometric mean to normalize *IDH-A* expression data. As a result, the stably expressed *IDH-A* acquired a diurnal fashion for accessions Titicaca and CHEN-109 (Fig 7B and 7C), showing that the use of unsuitable reference genes can misrepresent a non-diurnal gene as diurnally regulated. Furthermore, once an

appropriate reference has been found, it is possible to show the cyclic behavior of the candidate reference genes by normalizing the data of all candidates against the geometric mean of *IDH-A* and *PTB* (S9 Fig). In this way, an evidence that *IDH-A* and *PTB* are not diurnally regulated is also provided.

Interestingly, *ACT* has been validated as a suitable reference for diurnal expression studies in *Chenopodium ficifolium* [39]. In our study, *ACT* showed to be the best reference according to NormFinder, but this result was not supported by geNorm nor NormFinder methods. Moreover, important differences were observed between the expected pattern of the target genes, given by the simulation, and the obtained pattern by normalization with *ACT*, showing that there is not a unique reference gene that can be used across species. Additionally, in the *Chenopodium ficifolium* study only geNorm was used to test stability and one single gene was selected as a reference. It has been shown that to obtain measures of expression levels accurately, normalization by multiple housekeeping genes instead of one is required [7]. The reason for this is that a normalization against a factor (as it is the geometric mean of two or more reference genes) can effectively remove possible outlying values and abundance differences between the different genes. *CqIDH-A* and *CqPTB* were suitable as reference genes and their geometric mean was appropriate to normalize the diurnal expression of *CqBBX19*, *CqBTC1* and *CqGI*. *IDH* encodes for the isocitrate dehydrogenase enzyme, whereas the *PTB* gene product is the Polypyrimidine Tract Binding protein involved in pre-mRNA processing. An *IDH* ortholog in *B. vulgaris*, another member of the Amaranthaceae family, was validated as suitable reference gene in the expression analyses of two paralogs of the *FLOWERING LOCUS T* (*FT*) [24] and *PBT* was used as a reference gene in *C. quinoa* to normalize the expression of saponin biosynthesis genes after methyl jasmonate treatment [29]. In beet, *BBX19* acts together with *BTC1* in the circadian regulation of the flowering pathway [20]. Considering the close phylogenetic relationship between *B. vulgaris* and *C. quinoa*, *CqBBX19* and *CqBTC1* might similarly regulate the flowering time in quinoa through photoperiod pathway. *CqBTC1* diurnal expression pattern is comparable to the one given by for biennial vernalized *B. vulgaris* plants, since both expressions share a morning peak of expression [25]. *CqBBX19* circadian pattern of CHEN-109 is similar to the one obtained in sugar beet [20]. The *CqBTC1* and *CqBBX19* expression patterns in CHEN-109, which flowers later than Titicaca, suggest independent activity of these genes. On the other hand, the common expression pattern in Titicaca suggests a joint activity of *CqBTC1* and *CqBBX19*: these two genes clearly share the expression pattern from ZT-4 to ZT-12. *CqGI* peaks at ZT-8 for PI-587173 accession and follows a circadian rhythm similar to the one found for *A. thaliana* [26] and sugar beet [25]. In *A. thaliana*, *GI* is a key regulator of photoperiodic flowering [26]. However, sugar beet and *Arabidopsis* are considered long-day species and our expression experiments were conducted under short-days; thus, further experiments considering long day accessions investigated under long-day conditions are required to elucidate the putative role of *GI* in flowering time regulation of quinoa.

As the conclusion of this study, we were able to find a suitable reference gene combination, *CqIDH-A* and *CqPTB*, for the normalization of our diurnal expression data. Our results showed that using different reference genes for normalization, including those top ranked by stability, causes significant differences among the diurnal expression patterns of *CqBBX19* and *CqBTC1* for two different accessions of *C. quinoa* under short day conditions. This reinforces the fact that a more meticulous validation of reference genes is required for diurnal studies. Thus, we recommend the use of the presented comparison-based approach along with the use of the algorithms. The use of the comparison-based approach is recommended to disregard reference genes that greatly distort the expected expression pattern. In the future, the stable reference genes obtained in this study will significantly facilitate diurnal expression studies in quinoa by improving the accuracy of RT-qPCR data normalization.

## Supporting information

**S1 Table. $A_{260}/A_{280}$ and $A_{260}/A_{230}$ ratios of RNA isolated from diurnal leaf samples of the Titicaca accession.**
(PDF)

**S2 Table. Reference gene candidates obtained from cross validation of literature survey and *C. quinoa* shoot and root transcriptome data [17].** Selected genes were stably expressed at least in two out of four stress conditions in soil- and hydroponic-grown plants.
(PDF)

**S3 Table. Minimum Information for Publication of Quantitative Real-Time PCR Experiments (MIQE): Checklist.**
(PDF)

**S1 Fig. RNA and cDNA quality control by electrophoresis.** (A) 1.5% agarose gel of RNA isolated from leaf samples of the Titicaca accession; lanes 1 to 3: ZT-0, lanes 4 to 6: ZT-4, lanes 7 to 9: ZT-8, lanes 10 to 1, lanes 13 to 15: 2: ZT-12, lanes 13 to 15: ZT-16, lanes 16 to 18: ZT-20, lanes 19 to 21: ZT-24. Putative bands for 28S and 18S rRNA are indicated by arrows. Agarose gel was run for 40 min at 100 V (cropped gel). (B) 2.0% agarose gel of PCR products of cDNA synthesized from RNA leaf samples of Titicaca accession; lanes 1 to 3: ZT-0, lanes 4 to 6: ZT-4, lanes 7 to 9: ZT-8, lanes 10 to 1, lanes 13 to 15: 2: ZT-12, lanes 13 to 15: ZT-16, lanes 16 to 18: ZT-20, lanes 19 to 21: ZT-24. M: 50 bp ladder, PC: positive control, NC: water. Agarose gel was run for 30 min at 100 V (cropped gel).
(TIF)

**S2 Fig. Standard curves of the selected candidate and target genes.** Dilution factors 1:40 (2x), 1:20, 1:10 and 3:20 were used.
(TIF)

**S3 Fig. Amplification of PCR products of expected sizes for each of the candidate genes.** Lane 1: *ACT*, lane 2: *GAPDH*, lane 3: *IDH-B*, lane 4: *IDH-A*, lane 5: *PTB*, lane 6: *TUB*, lane 7: *UBQ*, lane 8: *RAN-3*, NC: water, M: 50 bp ladder. 2.0% agarose gel was run for 40 min at 100 V.
(TIF)

**S4 Fig. Melting curves of the candidate genes.** The curves were obtained from three technical replicates of three biological replicates (21 diurnal/circadian samples) for (A) CHEN-109 and (B) Titicaca.
(TIF)

**S5 Fig. Verification of the primer combinations specificity by Sanger sequencing.** Sequencing results of (A) *ACT*, *IDH-A*, *IDH-B* (B) *PTB*, *UBQ*, *TUB* and *RAN-3*. Sequences correspond to Titicaca accession aligned to the reference sequence (accession PI 614886) (Jarvis et al., 2017). Primers are annotated and coverage is shown (captions on the right side).
Fw = sequenced with the forward primer, Rv = sequenced with the reverse primer.
(TIF)

**S6 Fig. Reference gene candidates' intergroup variation between Titicaca and CHEN-109.** Variation was determined by NormFinder. Error bars represent the intragroup variation.
(TIF)

**S7 Fig. Diurnal relative expression of *GIGANTEA* (*CqGI*).** Relative *CqGI* expression normalized against each candidate gene, the combination of the two best genes for normalization of

Titicaca determined by BestKeeper (*IDH-A + PTB*) and a constant Cq value of 20 (Simulation) are shown. Expression corresponds to the accession PI-587173. The bar at the top indicates light (empty box) and dark (filled box) phases. Error bars represent the SEM of three biological replicates.
(TIF)

**S8 Fig. Simulated expression of the target genes with different Cq values.** Target genes expression was normalized against a constant Cq value of (A) 16 (B) 18 (C) 22 (D) 24.
(TIF)

**S9 Fig. Diurnal relative expression of *ACT, IDH-A, IDH-B, PTB, UBQ, TUB, GAPDH* and *RAN-3*.** Relative expression normalized against *IDH-A* and *PTB* geometric mean. The bar at the top indicates light (empty box) and dark (filled box) phases. Error bars represent the SEM of three biological replicates.
(TIF)

**S1 Raw images.** Full-length gels corresponding to cropped gels in (A) S1A Fig and (B) S1B Fig. Lanes labeled with an "X" in (B) correspond to different primer combinations tests for cDNA amplification by standard PCR. M: 50 bp ladder.
(TIF)

# Acknowledgments

We thank Monika Bruisch for her support for conducting and sampling the climate chamber experiment. Sequence analyses were carried out by the Institute for Clinical Molecular Biology (IKMB, Kiel University). We thank Prof. Mark Tester (King Abdullah University of Science and Technology) for providing the seeds used in our experiment.

# Author Contributions

**Conceptualization:** Christian Jung, Nazgol Emrani.

**Data curation:** Nathaly Maldonado-Taipe.

**Formal analysis:** Nathaly Maldonado-Taipe, Dilan S. R. Patirange.

**Investigation:** Nathaly Maldonado-Taipe, Dilan S. R. Patirange, Sandra M. Schmöckel.

**Methodology:** Nathaly Maldonado-Taipe.

**Project administration:** Christian Jung, Nazgol Emrani.

**Supervision:** Christian Jung, Nazgol Emrani.

**Writing – original draft:** Nathaly Maldonado-Taipe.

**Writing – review & editing:** Nathaly Maldonado-Taipe, Dilan S. R. Patirange, Sandra M. Schmöckel, Christian Jung, Nazgol Emrani.

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
