## [Decision Letter · Decision Letter 0]

2 Oct 2020

PONE-D-20-14035

Validation of suitable genes for normalization of diurnal gene expression studies in Chenopodium quinoa

PLOS ONE

Dear Dr. Nazgol Emrani,

Thank you for submitting your manuscript to PLOS ONE. After careful consideration, we feel that it has merit but does not fully meet PLOS ONE’s publication criteria as it currently stands. Therefore, we invite you to submit a revised version of the manuscript that addresses the points raised during the review process.

Your manusript has been seen by three reviewers. All three identified issues with design of the study and to various extent with methods and their description and support of conclusions by the data. Since the publications criteria #3 and #4 are not met your are required address this issues and ammend experiments. Of particular concern is the timing of taking samples and comparisions made, genes slected to be included in the study, and confusing, inconcise description of Figures amd insufficient support/arguments for dismissing reference genes.

We look forward to receiving your revised manuscript.

Kind regards,

Christian Schönbach, Dr.rer.nat.

Academic Editor

PLOS ONE

Journal Requirements:

Reviewers' comments:

Reviewer's Responses to Questions

**Comments to the Author**

1. Is the manuscript technically sound, and do the data support the conclusions?

Reviewer #1: Partly

Reviewer #2: Yes

Reviewer #3: Partly

2. Has the statistical analysis been performed appropriately and rigorously? 

Reviewer #1: Yes

Reviewer #2: Yes

Reviewer #3: Yes

3. Have the authors made all data underlying the findings in their manuscript fully available?

Reviewer #1: Yes

Reviewer #2: Yes

Reviewer #3: Yes

4. Is the manuscript presented in an intelligible fashion and written in standard English?

Reviewer #1: Yes

Reviewer #2: Yes

Reviewer #3: Yes

5. Review Comments to the Author

Reviewer #1: The manuscript entitled: “Validation of suitable genes for normalization of diurnal gene expression studies in Chenopodium quinoa” use three statistical algorithms (BestKeeper, geNorm and NormFinder) to validate potential candidate genes for normalization of diurnal expression studies in C. quinoa.

Mayor concerns:

1) In my opinion and after going thoroughly to the manuscript I do not find any biological question that the authors want to address. Even when the tittle of the manuscript is related to diurnal gene expression no conclusions, no discussion are given related to any diurnal expression study.

2) Rather than compared two different ecotypes, even when they have different time period of flowering they should do some comparison between day and night (circadian clock).

3) The authors choose, in their word three genes that represent the Amaranthaceae family, my concern is what have to do the rice there? Neither an explanation nor argument in discussion.

4) Also they analyze and made conclusion based on few publications to conclude that it is a problem with normalization. So my question is what happens with many publications related to transcriptome analysis that used qPCR for gene expression?

5) In my opinion the authors of this manuscript performed a nice statistical experiments and interpretation of data for their genes that it is a plus, however it will be better to go to a Journal of plant methods. Again, no biological address is discuss or presented.

Minor concern:

1) For this reviewer it was impossible to find the Danish accession, as the authors must be aware “Titicaca” is a name of a lake between Peru and Bolivia where a lot of quinoa is grown. May be the accession also is from Peru. In these days is very important to have very clear from where the accession is coming.

2) The authors also write Danish accession or Titicaca accession, it must be one way.

Reviewer #2: The manuscript “Validation of suitable genes for normalization of diurnal gene expression studies in Chenopodium quinoa” represents a very useful study aimed to find the most valid reference gene(s) which could be used in the study of circadian rhythmic gene expression in the important crop quinoa. The selection of appropriate reference gene(s) with a maxiumum expression stability is a crucial prerequisite for a successful study of gene expression. The authors used several programs in standard use such as NormFinder and GeNorm, utilized in silico transcriptomic resources as well as careful experimental validations. The topic belongs to the range of articles published in PLoS one. The manuscript is well written and easy to follow.

I have only a few comments.

1. Please, can you describe the simulation of the BBX expression in more detail? I could not find, how it was performed.

2. It is useful to design the primers “across” introns, to exclude amplification of the traces of gDNA. The quinoa genome is available, it would be useful to add this information.

3. Not only the choice of the gene, but also the selection of primers, are important. Can you briefly discuss the possible influence of primer choice (e.g. PCR efficiency) on the results?

4. The reference gene to normalize circadian geen expression experiment in the close relative of quinoa (C. ficifolium) has been recently published in Planta (250; 2019). Can you please compare your results with those? Actin11 was found as the best single reference also in your study. How big is the advantage to use two references compared to a single reference gene?

Reviewer #3: 1. Harvesting time points have been falsely mentioned in Line 79 as ‘every two hours’ as against every four hours that was actually done.

2. Line 206 is erroneous. Needs to be rectified.

3. The authors have used just one target gene, which is insufficient, and that too has not been introduced properly in terms of its cyclic expression pattern. The expected time of its peak has not been mentioned when the gene was first introduced in the text, so that the downstream analysis could be better understood. In an ideal study, four to five well known diurnal genes should have been taken, the peaks of which would be distributed to different parts of the day, most importantly dawn as well as dusk. Also, it is good to include genes with sharp (such as LHY, CCA1 having high expression) as well as shallow peaks (such as TOC1 having lower expression) so that the detrimental effect of the reference genes could be best understood.

4. The authors have used a new concept of simulation in RT-qPCR analysis that uses an arbitrary Cp value of 20 across all samples, in place of a reference gene, for normalizing the target genes. This concept somehow does not serve the very purpose of a reference gene whose main job is to minimize sample to sample variations arising in a sample set, due to human errors in pipetting and minor differences in RNA/ cDNA qualities and quantities. The concept needs to be supported by citing examples from any published literature if it exists.

5. The authors have taken just seven samples for sensitive diurnal studies and that too only for one day. Normally, a bi-hourly sampling should be pursued for at least two consecutive days, which should give similar expression pattern across both days. Since they have made a 4-hourly sampling for only one day, it becomes all the more important to check for the sample quality by testing expression of known diurnal genes like LHY, TOC, CCA1, etc. This will also eliminate the need to use simulation studies, as done by the authors.

6. Fig. 6 is not clear. In the figure legend it is mentioned that the expression of IDH-A with respect to other reference genes is depicted but the graphs are confusing to interpret.

7. Fig. 5 has been inadequately explained and vague interpretations have been drawn which makes it difficult to understand what the authors are trying to point out. How did the authors reach the conclusion that IDH-A and PTB genes are the best reference genes, without proving that they do not cycle, themselves.

8. A reference gene can be deemed unfit for diurnal experiments if it itself shows cycling behaviour or if its expression is very high or low, all of which can disturb the peaks of the target genes. The authors must clearly state the reason for rejecting some reference genes, while accepting others, by normalizing the data of all the reference genes chosen against the geometric mean of the best two or three reference genes. This will bring out the cyclic expression profile of the reference genes, if present. Doing this will certainly give more weightage to the claims made by the authors, as to why some genes can be used but the others can not to normalize diurnal datasets in Quinoa.

6. PLOS authors have the option to publish the peer review history of their article (what does this mean?). If published, this will include your full peer review and any attached files.

Reviewer #1: No

Reviewer #2: No

Reviewer #3: No

---

## [Author Response · Author response to Decision Letter 0]

10 Dec 2020

Dear reviewers:

We thank you for their valuable comments, which will contribute to the improvement of the manuscript. We have carefully revised the manuscript accordingly. In the following, we provide a point-by-point response to the reviewer’s comments. To clarify the points raised by the reviewers, we conducted more experiments and provided four additional figures (Figures 6, S7, S8 and S9). The changes to the manuscript are highlighted with track changes in the main text.

Please, note that all line numbers indicated in responses to reviewers’ comments are given according to the line numbers in the marked-up version of the manuscript including the track changes. Moreover, all the citations are presented according to the journal requirements in the main version of the manuscript (without track-changes). 

We hope that we could elucidate all the points raised by the reviewers and that the manuscript can be accepted after this revision.

Kind regards,

Nazgol Emrani 

Plant Breeding Institute

Christian-Albrechts-University of Kiel 

Am Botanischen Garten 1-9

D-24118 Kiel, Germany

Tel.: +49-4318802016

Fax: +49-4318802566

Email: n.emrani@plantbreeding.uni-kiel.de

Reviewer #1: The manuscript entitled: “Validation of suitable genes for normalization of diurnal gene expression studies in Chenopodium quinoa” use three statistical algorithms (BestKeeper, geNorm and NormFinder) to validate potential candidate genes for normalization of diurnal expression studies in C. quinoa.

Mayor concerns:

1) In my opinion and after going thoroughly to the manuscript I do not find any biological question that the authors want to address. Even when the tittle of the manuscript is related to diurnal gene expression no conclusions, no discussion are given related to any diurnal expression study.

Answer: 

The biological question is pronounced in the title “Validation of suitable genes for normalization” and covered in the introduction (lines 32 to 36). In order to clarify the concern of the reviewer lines 62 to 65 have been added/modified.

2) Rather than compared two different ecotypes, even when they have different time period of flowering they should do some comparison between day and night (circadian clock).

Answer:

The variation between day and night of our reference candidate genes can be already observed in Fig 2 and it is mentioned in lines 239 to 244. 

Although the main aim of the paper is to find a suitable reference to normalize RT-qPCR data and not to discuss the role of our target genes in the quinoa flowering pathway, we have extended our discussion about this topic in lines 475 to 493 to address the reviewer’s concern.

3) The authors choose, in their word three genes that represent the Amaranthaceae family, my concern is what have to do the rice there? Neither an explanation nor argument in discussion.

Answer:

The citation of the rice paper (Jain et al, 2018) was an important part of our literature survey and that is the reason to be cited in line 123 together with the Amaranthacea references. We are now aware of the confusion generated by this citation and have removed it from lines 123 and 217.

The rice paper is already included in our discussion (line 345, 349, 367, 436, 438). This paper is important for us not because of the crop (rice) but because it presents similar results regarding the validation of reference genes for diurnal studies, showing that some of the commonly used reference genes might depict a diurnal pattern (line 436-438).

4) Also they analyze and made conclusion based on few publications to conclude that it is a problem with normalization. So my question is what happens with many publications related to transcriptome analysis that used qPCR for gene expression?

Answer: 

To look for suitable reference genes for gene expression studies is a highly important task, which must be done before presenting any kind of RT-qPCR data. Unfortunately, many RT-qPCR studies still dismiss the seriousness of this step and ignore many publications (some of them cited in our paper), which refer to the importance of validating reference genes. 

5) In my opinion the authors of this manuscript performed a nice statistical experiments and interpretation of data for their genes that it is a plus, however it will be better to go to a Journal of plant methods. Again, no biological address is discuss or presented.

Answer: 

Several recent publications in Plos one address the question of finding accurate reference genes, considering its importance for reporting RT-qPCR data (Su et al., 2020; Ma et al., 2020).

To diminish the impression that our research is a “method” paper, we have modified lines 332-336, 397-399, 494-497. Moreover, although our paper’s aim is not to discuss the flowering pathway in quinoa, we have already given a first insight into this topic and provided data that can be used by the scientific community.

Minor concern:

1) For this reviewer it was impossible to find the Danish accession, as the authors must be aware “Titicaca” is a name of a lake between Peru and Bolivia where a lot of quinoa is grown. May be the accession also is from Peru. In these days is very important to have very clear from where the accession is coming.

Answer:

The reviewer may refer to https://www.tystofte.dk/en/varieties-status/national-list/, where it is possible to find that Puno, Titicaca and Vikinga are Danish varieties. 

2) The authors also write Danish accession or Titicaca accession, it must be one way.

Answer:

Following the reviewer’s recommendation, we have changed the name “Danish accession” to “Titicaca” in the manuscript

References:

Jain N, Vergish S, Khurana JP. Validation of house-keeping genes for normalization of gene expression data during diurnal/circadian studies in rice by RT-qPCR. Sci Rep. 2018;8(1):3203. doi: 10.1038/s41598-018-21374-1.

Su, X., Lu, L., Li, Y., Zhen, C., Hu, G., Jiang, K., ... & Chen, X. (2020). Reference gene selection for quantitative real-time PCR (qRT-PCR) expression analysis in Galium aparine L. Plos one, 15(2), e0226668.

Ma, L., Wu, J., Qi, W., Coulter, J. A., Fang, Y., Li, X., ... & Sun, W. (2020). Screening and verification of reference genes for analysis of gene expression in winter rapeseed (Brassica rapa L.) under abiotic stress. PloS one, 15(9), e0236577.

 

Reviewer #2: The manuscript “Validation of suitable genes for normalization of diurnal gene expression studies in Chenopodium quinoa” represents a very useful study aimed to find the most valid reference gene(s) which could be used in the study of circadian rhythmic gene expression in the important crop quinoa. The selection of appropriate reference gene(s) with a maxiumum expression stability is a crucial prerequisite for a successful study of gene expression. The authors used several programs in standard use such as NormFinder and GeNorm, utilized in silico transcriptomic resources as well as careful experimental validations. The topic belongs to the range of articles published in PLoS one. The manuscript is well written and easy to follow.

I have only a few comments.

1. Please, can you describe the simulation of the BBX expression in more detail? I could not find, how it was performed.

Answer:

We have added lines 195 to 197 to clarify this point. The simulation procedure now is described from line 190 to 205.

2. It is useful to design the primers “across” introns, to exclude amplification of the traces of gDNA. The quinoa genome is available, it would be useful to add this information.

Answer:

We considered this point during our primer design and have modified the text accordingly (lines 137-139). Moreover, we marked in Table 1 the pair of primers which are designed across introns. 

3. Not only the choice of the gene, but also the selection of primers, are important. Can you briefly discuss the possible influence of primer choice (e.g. PCR efficiency) on the results?

Answer:

We added lines 374 to 383 to discuss this point.

4. The reference gene to normalize circadian geen expression experiment in the close relative of quinoa (C. ficifolium) has been recently published in Planta (250; 2019). Can you please compare your results with those? Actin11 was found as the best single reference also in your study. How big is the advantage to use two references compared to a single reference gene?

Answer:

The reviewer’s concern is addressed in the added lines 458-468.

 

Reviewer #3: 

1. Harvesting time points have been falsely mentioned in Line 79 as ‘every two hours’ as against every four hours that was actually done.

Answer:

The harvesting time points were indeed ‘every two hours’; nevertheless, we considered only the ZT points mentioned in line 99 due to the time constraints and the aim of our experiment. To avoid confusion, we added line 97-98.

2. Line 206 is erroneous. Needs to be rectified.

Answer: the line has been corrected (line 230-231).

3. The authors have used just one target gene, which is insufficient, and that too has not been introduced properly in terms of its cyclic expression pattern. The expected time of its peak has not been mentioned when the gene was first introduced in the text, so that the downstream analysis could be better understood. In an ideal study, four to five well known diurnal genes should have been taken, the peaks of which would be distributed to different parts of the day, most importantly dawn as well as dusk. Also, it is good to include genes with sharp (such as LHY, CCA1 having high expression) as well as shallow peaks (such as TOC1 having lower expression) so that the detrimental effect of the reference genes could be best understood.

Answer:

We have added a complete analysis of a new target gene BOLTING TIME CONTROL 1 (BTC1) for two accessions and an extra analysis of GIGANTEA (GI) for a third accession. Thus, fulfilling the requirements of the reviewer considering a gene that is expected to peak at dawn (with a shallow peak; BBX19), a gene that peaks at dusk (BTC1) and a gene with a sharp peak (GI). The expected times of the peaks have been added to lines 207 to 209. The text has been modified in all parts where it was affected by the addition of the new target genes.

4. The authors have used a new concept of simulation in RT-qPCR analysis that uses an arbitrary Cp value of 20 across all samples, in place of a reference gene, for normalizing the target genes. This concept somehow does not serve the very purpose of a reference gene whose main job is to minimize sample to sample variations arising in a sample set, due to human errors in pipetting and minor differences in RNA/ cDNA qualities and quantities. The concept needs to be supported by citing examples from any published literature if it exists.

Answer:

The selected Cq value will not alter the outcome of the simulated pattern since the by 2^(-∆∆Cq) reports the expression as relative values; thus, the differential value ∆∆Cq will produce the same pattern as long as the reference value is constant among biological replicates and sampling points. To show that varying the selected Cq value for the simulation does not affect the simulation outcome, we have added S8 Fig. The concept can be supported by the definitions of the basic principles of a reference gene (added lines 408 to 414). As far as we know, no other paper reports a simulation as we have performed, being thus, part of the novelty of our paper.

5. The authors have taken just seven samples for sensitive diurnal studies and that too only for one day. Normally, a bi-hourly sampling should be pursued for at least two consecutive days, which should give similar expression pattern across both days. Since they have made a 4-hourly sampling for only one day, it becomes all the more important to check for the sample quality by testing expression of known diurnal genes like LHY, TOC, CCA1, etc. This will also eliminate the need to use simulation studies, as done by the authors.

Answer:

Taking the words of the reviewer, we expect that the addition of BTC1 and GI, as “known diurnal genes”, enhances the presented results. Nevertheless, we show the addition of these genes does not eliminate the need of simulation. Conversely, the addition of target genes supports the point that reference genes have to be carefully chosen, not relying exclusively on the widely used algorithms (Fig 6 and S7 Fig).

We agree with the reviewer that in the “ideal case”, samples should be taken in a course of a few days. Nevertheless, relevant publications related to our study, which show BBX19, BTC1 (Dally et al.,2006; Vogt et al., 2014) and GI diurnal expressions, have been performed during the course of a day. In the case of GI in four-hour intervals (David et al.,2006). Moreover, several papers, which make even stronger statements and conclusions from diurnal studies than those presented in our study, have been performed during the course of a day (Na et al., 2020; Zhang et al., 2020). Finally, we believe that the addition of an extra sampling day should not be indispensable because we present variation of a considerable number of genes (putative references and target genes) in three different accessions. 

6. Fig. 6 is not clear. In the figure legend it is mentioned that the expression of IDH-A with respect to other reference genes is depicted but the graphs are confusing to interpret.

Answer:

Fig 6., called now Fig 7, has been modified for a better understanding. The aim of this figure is to show that using inadequate reference genes can result in representing a non-diurnal gene as diurnally regulated. For a better explanation lines 438-447 have been added/modified.

7. Fig. 5 has been inadequately explained and vague interpretations have been drawn which makes it difficult to understand what the authors are trying to point out. How did the authors reach the conclusion that IDH-A and PTB genes are the best reference genes, without proving that they do not cycle themselves.

Answer:

To reach this conclusion we used two complementing analyses:

 The results of GeNorm, which indicate IDH-A and PTB as the best references and point that the geometric mean of two reference genes is required for accurate normalization.

 Since the stability results among the three software were not the same, another way of evaluation is required. Here we used the simulation to discard unsuitable reference genes according to the parameters described in the method section from line 199 to 205. 

To clarify how we reached the main conclusion we added/modified lines 285-287, 290-291, 305-310. Together with this, we modified the interpretations of Figure 5 and clarified why we discarded some genes in lines 293-294. We also added an example of how we discarded genes related to the new Fig 6, which depicts results for BTC1 (lines 301-304).

The three software analyses intend to prove that the candidates do not show a diurnal pattern themselves. Nevertheless, we have shown that the results of these software are not enough evidence of stable expression during the course of a day. As an evidence that the genes do not cycle themselves, we added S9 fig, which in fact, was created following the directions of the reviewer at point 8. 

8. A reference gene can be deemed unfit for diurnal experiments if it itself shows cycling behaviour or if its expression is very high or low, all of which can disturb the peaks of the target genes. The authors must clearly state the reason for rejecting some reference genes, while accepting others, by normalizing the data of all the reference genes chosen against the geometric mean of the best two or three reference genes. This will bring out the cyclic expression profile of the reference genes, if present. Doing this will certainly give more weightage to the claims made by the authors, as to why some genes can be used but the others can not to normalize diurnal datasets in Quinoa.

Answer:

We thank the reviewer for this suggestion, which will certainly reinforce the presented results. We have added S9 fig and modified the text in lines 447 to 450. 

References:

Dally, N., Eckel, M., Batschauer, A., Höft, N., & Jung, C. (2018). Two CONSTANS-LIKE genes jointly control flowering time in beet. Scientific reports, 8(1), 1-10.

David, K. M., Armbruster, U., Tama, N., & Putterill, J. (2006). Arabidopsis GIGANTEA protein is post-transcriptionally regulated by light and dark. FEBS letters, 580(5), 1193-1197.

Ng, J. W. X., Tan, Q. W., Ferrari, C., & Mutwil, M. (2020). Diurnal. plant. tools: comparative transcriptomic and co-expression analyses of diurnal gene expression of the Archaeplastida kingdom. Plant and Cell Physiology, 61(1), 212-220.

Vogt, S. H., Weyens, G., Lefèbvre, M., Bork, B., Schechert, A., & Müller, A. E. (2014). The FLC-like gene BvFL1 is not a major regulator of vernalization response in biennial beets. Frontiers in plant science, 5, 146.

Zhang, X., Fatima, M., Zhou, P., Ma, Q., & Ming, R. (2020). Analysis of MADS-box genes revealed modified flowering gene network and diurnal expression in pineapple. BMC genomics, 21(1), 1-16.

---

## [Decision Letter · Decision Letter 1]

14 Jan 2021

PONE-D-20-14035R1

Validation of suitable genes for normalization of diurnal gene expression studies in Chenopodium quinoa

PLOS ONE

Dear Dr. Emrani,

Thank you for submitting your manuscript to PLOS ONE. After careful consideration, we feel that it has merit but does not fully meet PLOS ONE’s publication criteria as it currently stands. Therefore, we invite you to submit a revised version of the manuscript that addresses the points raised during the review process.

The revised version addressed most of the the previous concerns and suggestion. Reviewer #2 still has reservations regarding the assumption of constant Cq value 20 across all samples in context potentially varying quantities (tube to tube differences) after DNAse treatment, reverse transcription (after Nanodrop) and the Nanodrop measurement itself. Although MIQUE guidelines state "...it is advisable to measure all samples with a single method only and to report this information" they also state "The preferred method for quantifying RNA uses fluorescent RNA-binding dyes (e.g., RiboGreen), which are best for detecting low target concentrations." This needs to be addressed in light of publication criterion 3.

Aside from the reviewer comments you are encouraged to use the MIQUE checklist (https://rdml.org/files/docs/MIQE_checklist.xls) and include it as supplement table.

We look forward to receiving your revised manuscript.

Kind regards,

Christian Schönbach, Dr.rer.nat.

Academic Editor

PLOS ONE

Reviewers' comments:

Reviewer's Responses to Questions

**Comments to the Author**

1. If the authors have adequately addressed your comments raised in a previous round of review and you feel that this manuscript is now acceptable for publication, you may indicate that here to bypass the “Comments to the Author” section, enter your conflict of interest statement in the “Confidential to Editor” section, and submit your "Accept" recommendation.

Reviewer #1: All comments have been addressed

Reviewer #2: (No Response)

2. Is the manuscript technically sound, and do the data support the conclusions?

Reviewer #1: Partly

Reviewer #2: Partly

3. Has the statistical analysis been performed appropriately and rigorously? 

Reviewer #1: Yes

Reviewer #2: Yes

4. Have the authors made all data underlying the findings in their manuscript fully available?

Reviewer #1: Yes

Reviewer #2: Yes

5. Is the manuscript presented in an intelligible fashion and written in standard English?

Reviewer #1: Yes

Reviewer #2: Yes

6. Review Comments to the Author

Reviewer #1: I would like to thanks the authors for answer my concerns and clarify most of my points. However still I think that represent more a methodology rather than address a biological question. Based on that I will leave the decision to accept/reject the manuscript in the hands of the Editor.

Reviewer #2: The authors of the manuscript “Validation of suitable genes for normalization of diurnal gene expression studies in Chenopodium quinoa“ improved their article substantially in the course of revision and clarified most issues.

However, one of my concerns persists. It is the simulation method, when the authors took the constant Cp value 20 across all the samples. This approach would work supposing that exactly identical amount of cDNA was added to each qPCR reaction. The authors relied on Nanodrop measurements of RNAs before performing reversion transcription and DNAse treatment. However, DNAse treatment and reverse transcription may generate tube-to-tube differences in the final cDNA yield. Thus, I do not consider this simulation to be an appropriate tool to verify reference gene stability. Fluorescence-based estimation of cDNA amount shall be used instead to measure equality of input to RT PCR reaction. Please, consult the paper Libus et al., Biotechniques 2006.

7. PLOS authors have the option to publish the peer review history of their article (what does this mean?). If published, this will include your full peer review and any attached files.

Reviewer #1: No

Reviewer #2: No

---

## [Author Response · Author response to Decision Letter 1]

10 Feb 2021

Dear Editor, 

Thank you for your e-mail enclosing the reviewer’s comments. We thank the reviewers for their valuable comments, which will contribute to the improvement of the manuscript.

We have carefully revised the manuscript accordingly. In the following, we provide a point-by-point response to the reviewer’s comments. To clarify the concern of reviewer 2 and as encouraged by the Editor, we modified the Discussion and provided one additional table (Table S3). The changes to the manuscript are highlighted with track changes in the main text.

Please, note that all line numbers indicated in responses to reviewers’ comments are given according to the line numbers in the marked-up version of the manuscript including the track changes. Moreover, all the citations are presented according to the journal requirements in the main version of the manuscript (without track-changes). All authors have received the reviews and been informed about the re-submission of the manuscript.

This study was partially funded by the Competitive Research Grant of the King Abdullah University of Science and Technology, Saudi Arabia awarded to CJ (Grant number: OSR-2016-CRG5-2966-02). The funders had no role in study design, data collection and analysis, decision to publish, or preparation of the manuscript. There was no additional external funding received for this study. The rest of the costs were covered internally by the institute budget of the Plant Breeding Institute, Kiel University.

We hope that we could elucidate all the points raised by the reviewers and that the manuscript can be accepted after this revision.

Kind regards,

Nazgol Emrani 

Plant Breeding Institute

Christian-Albrechts-University of Kiel 

Am Botanischen Garten 1-9

D-24118 Kiel, Germany

Tel.: +49-4318802016

Fax: +49-4318802566

Email: n.emrani@plantbreeding.uni-kiel.de

Response to reviewers

Reviewer #1: I would like to thanks the authors for answer my concerns and clarify most of my points. However still I think that represent more a methodology rather than address a biological question. Based on that I will leave the decision to accept/reject the manuscript in the hands of the Editor.

Answer:

We thank the reviewer for spending the time to review the manuscript again. We feel like we have addressed this point in the last revision under Introduction lines 31 to 35 and 61 to 64.

Reviewer #2: The authors of the manuscript "Validation of suitable genes for normalization of diurnal gene expression studies in Chenopodium quinoa" improved their article substantially in the course of revision and clarified most issues.

However, one of my concerns persists. It is the simulation method, when the authors took the constant Cp value 20 across all the samples. This approach would work supposing that exactly identical amount of cDNA was added to each qPCR reaction. The authors relied on Nanodrop measurements of RNAs before performing reversion transcription and DNAse treatment. However, DNAse treatment and reverse transcription may generate tube-to-tube differences in the final cDNA yield. Thus, I do not consider this simulation to be an appropriate tool to verify reference gene stability. Fluorescence-based estimation of cDNA amount shall be used instead to measure equality of input to RT PCR reaction. Please, consult the paper Libus et al., Biotechniques 2006.

Answer:

We thank the reviewer for their comments. We agree that fluorescent methods may be better suited to quantify RNA or to estimate the amount of cDNA, especially when different amounts of RNA and/or contaminants can be expected, for instance because different tissues were used, or strong stresses were imposed on the plants. In this study, we focused on one tissue only (leaves) and plants were all at the same developmental stage. Moreover, the Nanodrop-spectra suggest that samples are similar in purity. Nevertheless, we have discussed the drawbacks in lines 396-404 and 469-471. We have also cited the mentioned reference as an advisable method for quantification of cDNA (line 404). The use of the simulation is intended to work together with the widely used methods to, mainly, disregard the reference genes that produce a highly distorted pattern from the “expected” one (already mentioned in line 389). We have followed the MIQE guidelines, which allows the use of Nanodrop (added S3 Table and line 596).

---

## [Decision Letter · Decision Letter 2]

19 Feb 2021

Validation of suitable genes for normalization of diurnal gene expression studies in Chenopodium quinoa

PONE-D-20-14035R2

Dear Dr. Emrani,

We’re pleased to inform you that your manuscript has been judged scientifically suitable for publication and will be formally accepted for publication once it meets all outstanding technical requirements.

Kind regards,

Christian Schönbach, Dr.rer.nat.

Section Editor

PLOS ONE

Additional Editor Comments (optional):

Reviewers' comments:

Reviewer's Responses to Questions

**Comments to the Author**

1. If the authors have adequately addressed your comments raised in a previous round of review and you feel that this manuscript is now acceptable for publication, you may indicate that here to bypass the “Comments to the Author” section, enter your conflict of interest statement in the “Confidential to Editor” section, and submit your "Accept" recommendation.

Reviewer #2: All comments have been addressed

2. Is the manuscript technically sound, and do the data support the conclusions?

Reviewer #2: (No Response)

3. Has the statistical analysis been performed appropriately and rigorously? 

Reviewer #2: (No Response)

4. Have the authors made all data underlying the findings in their manuscript fully available?

Reviewer #2: (No Response)

5. Is the manuscript presented in an intelligible fashion and written in standard English?

Reviewer #2: (No Response)

6. Review Comments to the Author

Reviewer #2: (No Response)

7. PLOS authors have the option to publish the peer review history of their article (what does this mean?). If published, this will include your full peer review and any attached files.

Reviewer #2: No

---

## [Editor Report · Acceptance letter]

2 Mar 2021

PONE-D-20-14035R2 

Validation of suitable genes for normalization of diurnal gene expression studies in *Chenopodium quinoa*

Dear Dr. Emrani:

I'm pleased to inform you that your manuscript has been deemed suitable for publication in PLOS ONE. Congratulations! Your manuscript is now with our production department. 

Kind regards, 

on behalf of

Dr. Christian Schönbach 

Section Editor

PLOS ONE